# Development of Novel Polypropylene Syntactic Foams Containing Paraffin Microcapsules for Thermal Energy Storage Applications

**DOI:** 10.3390/molecules27238520

**Published:** 2022-12-03

**Authors:** Francesco Galvagnini, Andrea Dorigato, Luca Fambri, Alessandro Pegoretti

**Affiliations:** Department of Industrial Engineering and INSTM Research Unit, University of Trento, Via Sommarive 9, 38123 Trento, Italy

**Keywords:** syntactic foams, polypropylene, hollow glass microspheres, thermal energy storage, phase change materials, thermal properties, mechanical properties

## Abstract

polypropylene (PP) syntactic foams (SFs) containing hollow glass microspheres (HGMs) possess low density and elevated mechanical properties, which can be tuned according to the specific application. A possible way to improve their multifunctionality could be the incorporation of organic Phase Change Materials (PCMs), widely used for thermal energy storage (TES) applications. In the present work, a PCM constituted by encapsulated paraffin, having a melting temperature of 57 °C, was embedded in a compatibilized polypropylene SF by melt compounding and hot pressing at different relative amounts. The rheological, morphological, thermal, and mechanical properties of the prepared materials were systematically investigated. Rheological properties in the molten state were strongly affected by the introduction of both PCMs and HGMs. As expected, the introduction of HGMs reduced both the foam density and thermal conductivity, while the enthalpy of fusion (representing the TES capability) was proportional to the PCM concentration. The mechanical properties of these foams were improved by the incorporation of HGMs, while they were reduced by addition of PCMs. Therefore, the combination of PCMs and HGMs in a PP matrix generated multifunctional materials with tunable thermo-mechanical properties, with a wide range of applications in the automotive, oil, textile, electronics, and aerospace fields.

## 1. Introduction

Composites generally consist of two components, i.e., a matrix and an embedded filler, which are responsible for giving peculiar electrical, mechanical, thermal, and chemical properties. The choice of the materials to be combined is strictly dependent on the final application. In most cases, the focus is on the improvement of mechanical performance and also reducing the weight of the final components. This aspect is particularly important for non-stationary products, where a weight reduction can significantly reduce fuel consumption and consequently, greenhouse gas (GHG) emissions [1,2,3]. 

When lightness is a requirement, foams play an important role thanks to their high specific volume and specific strength [4]. A special class of foams to consider, especially when compressive strength is needed, is that of syntactic foams (SFs). Unlike traditional foams where porosity is formed during the foaming process, in syntactic foams the porosity can be obtained simply by incorporating pre-formed bubbles [5]. These hollow spheres can be found on the market in a large variety of materials and sizes. When incorporated into a matrix, they can reduce the overall density of the composite, even when they are made of glass (HGM). In this way, it is possible to reach closed cell porosity values of about 60% by volume, also reducing the final thermal conductivity of the system [6,7,8]. This class of foams found its first application in the marine field, thanks to their elevated chemical stability and buoyancy [9]. Syntactic foams were then applied in many other fields, such as the aerospace and automotive industries. Although the majority of syntactic foams produced today are constituted of epoxy/HGMs systems [5,8,10,11,12,13,14,15,16], thermoplastic-based syntactic foams are gaining increasing research interest, due to their easy recyclability at their end of life [5,17,18]. By varying the HGM diameter, size distribution, shell thickness, volume fraction, and surface reactivity, the final properties of the resulting SFs can be finely tuned. For instance, high content of low-density HGMs results in a light component with limited impact strength. It is possible to overcome this limitation by incorporating a third phase, such as glass or carbon fibers, nanoclays, or carbon nanotubes [14,19]. Moreover, the addition of conductive fillers can also impart additional properties to the foams, such as elevated thermal and electrical conductivity, thus creating multifunctional composites.

This property allows the storage and release of heat at specific temperature levels when and where necessary, and it can be thus exploited for passive thermal management applications [20,21,22]. Phase Change Materials (PCMs) are a subclass of TES systems that are based on solid/liquid phase transitions to store and release heat. By tailoring the PCM concentration and the working temperature, PCMs can limit the amplitude of temperature oscillations in different thermodynamic systems, thus reducing the need for active thermal management operations and lowering the overall energy consumption. They find application in various fields, such as electronics, textiles, batteries, and construction. For instance, some solar power plants use PCMs to store daily heat to allow electricity production overnight [23,24,25,26,27,28,29,30]. Most PCMs usually operate in a temperature range between −20 and 120 °C, and the class of organic PCMs is the most suitable one in this temperature interval. It comprises some organic oligomers like alcohols, fatty acids, poly (ethylene glycols) (PEGs), and paraffins. They are inexpensive, can store a significant amount of heat (up to 300 J/g), and their transition temperature can be easily tuned by simply changing their molecular weight. One of the main issues with these materials is related to their leakage in the molten state. To overcome this problem, several confinement strategies were developed. The simplest one is their confinement in polymeric micro- or nano-capsules [21,27,31,32]. This approach solves the leakage problem, avoids interactions with the matrix or with the surrounding environment, and stabilizes PCM volume variations. 

There are several examples in the literature of composite materials incorporating PCM capsules to impart TES features, and the attention of our group in this research field was mainly focused on paraffinic PCMs [28,33,34,35,36,37,38,39,40,41]. In particular, an epoxy-based syntactic foam containing different amounts of PCM microcapsules has been recently investigated. The lightening power and the thermal insulation capacity provided by HGMs were highlighted, while the PCM capsules imparted interesting TES properties (about 70 J/g with a paraffin content of 40% by volume). On the other hand, the mechanical performances were slightly reduced by the PCM addition [8,42]. Quite interestingly, no papers can be found in the open literature on the development of multifunctional thermoplastic foams with TES properties.

Therefore, in the present work a commercial-grade polypropylene, compatibilized with maleated PP, was used as a matrix for the development of new syntactic foams incorporating HGMs and an encapsulated PCM with a melting temperature of 57 °C. A comprehensive rheological, microstructural, thermal, and mechanical characterization of the resulting materials was then performed on ten different formulations. 

## 2. Results and Discussions

### 2.1. Rheological and Morphological Properties

A simple way to investigate the processability of thermoplastics is to measure their rheological properties. For this purpose, two different tests were conducted, frequency sweep, for determining G′ and G″ trends as a function of angular velocity (ω), and flow sweep, to detect the dependence of the viscosity (η) on the applied shear rate ($\dot{\gamma }$). More details regarding the test procedures and parameters are reported in the Materials and Methods section. Figure 1 and Table 1 show the results of the frequency sweep analysis conducted on the prepared foams. Figure 1 shows only the G′ and G″ trends of four representative compositions, H0-P0, H20-P20, H20-P0, and H0-P20, while Table 1 reports the crossover point coordinates of all the analyzed samples.

Figure 1 shows that for all the compositions under investigation, below the crossover frequency G″ is higher than G′, resulting in a dominance of the viscous over the elastic behavior. On the contrary, at frequencies higher than the crossover point G″ < G′, because the molten polymer macromolecules do not have enough time to relax the stress [43]. Considering compositions containing only PCM, from Table 1 it can be noticed that the presence of PCM capsules tends to shift the crossover point to higher frequencies with respect to the PP matrix (H0-P0). On the other hand, in compositions containing only HGM the crossover point is shifted to higher values of G′ and G″. As expected, by combining the two fillers their effects are also combined. For example, the frequency of the crossover point of H20-P20 is 47.5 rad/s, which is higher than that shown by both H20-P0 (26.7 rad/s) and H0-P20 (28.2 rad/s) samples. The crossover frequency of H20-P20 (47.5 rad/s) is the highest one, even when compared to mono-filler compositions with the same amount of filler, such as the H40-P0 sample (21.6 rad/s). These trends are well described in Figure 2, where the crossover frequency of the prepared foams is represented in a ternary diagram.

The results of the flow sweep rheological tests on seven representative compositions are reported in Figure 3.

Figure 3 highlights the non-Newtonian behavior of the analyzed foams, which is typical rheological behavior of polymeric materials [4]. It can be noticed that samples incorporating HGMs possess higher viscosity values compared to the PP matrix proportionally to their concentration, while the presence of PCM capsules shifts the curves at lower viscosity levels. This could be associated with the partial breakage of the PCM capsules during the production process, which implies very high shear stresses. In addition, during the mixing step the inner PCM is molten, therefore becoming free to leak from the broken capsules and thus acting as a sort of lubricating agent. When both fillers are present, the two effects tend to balance out. 

One of the most important reasons why syntactic foams are so interesting for marine environments is their tailorable density, coupled with high specific strength, stiffness, and buoyancy. Thus, a helium pycnometer was used to measure the density (ρ) of the prepared syntactic foams. The numerical results are reported in Table 2, while the ternary diagrams shown in Figure 4a,b represent the trends of the density and the void content.

As can be observed in Figure 4a, the introduction of HGMs reduces, proportionally to their volume fraction, the density of the composites. In fact, the density passes from 0.90 g/cm^3^ for the neat matrix up to 0.70 g/cm^3^ for the H40-P0 foam (−22%). Quite surprisingly, the PCM capsules also lead to a density decrease. The density of PCM capsules is 0.94 g/cm^3^, which is higher than that of the matrix, therefore in principle the density of the resulting composites should be increased. This behavior could be explained by the void development during the mixing stage generated by the partial evaporation of the PCM leaking from the broken capsules, acting as a sort of foaming agent, with a consequent density decrease. The worst situation can be observed in the H20-P20 foam, where the porosity reaches 10.2%, due to the additional tribological effect provided by the HGMs introduction, which increases the PCM capsule rupture.

To confirm the PCM capsule partial breakage during the compounding operations and to study the microstructure of the prepared foams, SEM analysis was conducted on the cryo-fractured surfaces of four selected compositions (H0-P0, H20-P20, H40-P0, and H0-P30), as shown in Figure 5a–d.

Figure 5a shows the typical fracture surface morphology of polypropylene, while the breakage of the PCM capsules can be clearly seen in Figure 5b (showing the H0-P30 foam). Most of the capsules are broken or deformed, supporting the hypothesis of void generation due to the leakage of the molten PCM (see Table 2). Even if they are partially broken it does not mean that the PCM leaves the composite completely. Indeed, this picture also shows that the matrix is capable of acting as a second shell, avoiding any further loss of PCM. This evidence will be confirmed by DSC results. Moreover, Figure 5b shows a poor interfacial adhesion between the PCM capsules and the PP matrix, while Figure 5c reveals that the silanization process of HGMs and the presence of a compatibilizer in the matrix lead to a good PP/HGM interfacial adhesion. This result could provide a beneficial effect on the elastic modulus and strength of the material, especially under compression. Figure 5d displays the fracture surface of the composition containing both HGM and PCM capsules, and it is confirmed that the presence of HGM beads further increases the breakage tendency of PCM capsules, due to the higher shear stresses developed during the compounding stage. Moreover, this result agrees with the fact that this composition is also the most porous one, as demonstrated in density measurements.

### 2.2. Thermal Properties

One of the simplest ways to measure TES capability of materials is through differential scanning calorimetry (DSC), which quantifies the phase change temperatures of PCM (T_m_ and T_c_), and its enthalpy of fusion and crystallization (ΔH_m_ and ΔH_c_). The specific heat has also been evaluated but, for the sake of brevity, the results will not be reported in this work. Figure 6a,b shows the thermogram of the first DSC scan of five representative compositions and the ternary diagram representing the enthalpy of fusion evaluated in the first heating scan, while Table 3 summarizes the results of the first heating and cooling DSC scan of all the compositions.

Some compositions do not show phase transitions due to the absence of PCM capsules, as reported in Table 3. For all the compositions containing PCM, Figure 6a shows an increase in the heat stored as the volume fraction of PCM increases. This proportionality can also be clearly seen in the ternary diagram in Figure 6b. The presence of two melting peaks in Figure 6a is due to the presence of paraffin fractions with different molecular weights. Moreover, these results also demonstrate that the PP matrix acts as a second shell, because most of the samples possess enthalpy of fusion close to the theoretical one (ΔH_m,th_), evaluated considering the PCM content in the foams. For example, H0-P30 has a melting enthalpy of 58.4 J/g, close to the ΔH_m,th_ value of 62.1 J/g. In the sample characterized by the highest porosity degree (i.e., H20-P20) the enthalpy of fusion differs only by 5% from the theoretical one. The discrepancy between melting and crystallization temperatures is due to the thermal inertia of the system and the undercooling phenomena, typical at this heating/cooling rate. By reducing the cooling rate, for example by 1 °C/min, it is possible to get the two temperatures closer [44]. 

Another method used in this work to investigate TES properties, is the temperature profiling during heating and cooling stages. Seven compositions were chosen to perform this test, H0-P0, H0-P20, H0-P30, H20-P0, H40-P0, H7-P7, and H20-P20. The temperatures of the samples were monitored through K-type thermocouples during heating from room temperature and 100 °C, and during cooling from 100 °C to room temperature. The heating step was conducted in a preheated oven, whereas the cooling step was performed outside the oven. In each step, the time required for the samples to reach a temperature difference between 35 and 80 °C was recorded (t_35–80_ and t_80–35_). The temperature profiles obtained both in the heating and cooling stages are shown in Figure 7, while Table 4 lists the results of these tests in terms of t_35–80_ and t_80–35_ values.

The presence of PCM clearly influences the temperature profiles, as shown in Figure 7, with the presence of an inflection at the melting point of PCM. This results in a longer time needed to reach the target temperature with respect to the PP matrix, as also reported in Table 4. In this sense, PCM is capable of giving TES properties to the material since it can store and release heat during temperature transients. Another interesting aspect highlighted by Figure 7 and Table 4, is that introduction of HGMs reduces the heating and cooling times. As reported in a technical paper produced by 3M^TM^ [45], syntactic foams show at the same time, reduced thermal conductivity and increased thermal diffusivity due to the presence of HGMs that reduce the final density (see Equation (3)). This results in a reduction of the transient time, a peculiarity that can be useful in injection molding to increase the production rate. 

Thermal conductivity is correlated with density, thermal diffusivity, and specific heat by Equation (3). This relationship was applied to the results obtained from the laser flash analyzer (LFA), which gives the thermal diffusivity, to determine the thermal conductivity of the analyzed samples. The tests were performed at three temperatures, 20, 45, and 70 °C, and the resulting thermal conductivity values are presented in Figure 8. 

As expected, the presence of HGMs significantly decreases the thermal conductivity of the samples, which is in agreement with the results reported in the previous work of our group [8]. Different to that work, in which the thermal conductivity was correlated only to the HGMs content, in this work the PCM content also plays an important role. In fact, the compositions that show the lowest values of thermal conductivity are the ones containing both HGM and PCM, i.e., H20-P20 and H27-P7 foams. For example, H40-P0, which contains only HGMs, shows a thermal conductivity at 20 °C of 0.190 W/m∙K, which is 29% lower than that of the neat matrix. When half of the HGMs are replaced with PCMs, like in H20-P20 foam, the thermal conductivity drops down to 0.162 W/m∙K (−15% than H40-P0, and −40% than the matrix). This behavior can be explained by the development of voids in the mixed compositions (containing both PCM and HGMs), that decrease the thermal conductivity and the density of the foams.

### 2.3. Mechanical Properties

In this work, all the prepared compositions were tested under quasi-static tensile conditions, to evaluate their elastic modulus (E_t_), stress and strain at yield (σ_y_ and ε_y_), and stress and strain at break (σ_B_ and ε_B_). The elastic modulus and the stress at the break of the samples were also normalized by their density (E_t_/ρ and σ_B_/ρ). Figure 9 reports the representative stress–strain curves of seven selected compositions, while Table 5 lists the most important tensile properties. Moreover, the elastic modulus, stress at break, and their normalized values are represented in ternary diagrams in Figure 10a–d.

Figure 9 evidences a neat drop in tensile strain of all the compositions containing fillers (PCM or HGMs) compared to the neat matrix. The compositions that better perform in this sense are those incorporating only PCM capsules, where H0-P30 shows a strain at break “only” 80% lower than the PP matrix. Similarly, samples containing only HGMs exhibit the worst situation, with a strain at break decrease of 84% compared to the matrix. Concerning the elastic modulus, Figure 9 reveals the benefits provided by HGMs, by increasing the stiffness from 1509 MPa of the matrix up to 2480 MPa of the H30-P0 foam (+64%). This is due to the stiff nature of HGMs. On the other hand, PCMs reduce the elastic modulus from 1509 MPa down to 758 MPa of H0-P30 foam (−50%), because of the limited stiffness of the PCM capsules. These trends are better represented in the ternary diagram of Figure 10a. This diagram also shows a proportionality of the elastic modulus when the PCMs are substituted by an equivalent volumetric amount of HGMs (by moving horizontally from right to left in the graph). The normalized elastic modulus, shown in Figure 10b, displays an analogous situation where Et/ρ proportionally changes with the PCM degree of substitution with HGMs. As for the elastic modulus, the stress at break is also influenced similarly. While introduction of HGMs leads to a substantial σ_B_ retention, the presence of PCMs lowers σ_B_ from 33.5 MPa of the matrix down to 15.5 MPa of the H0-P30 foam (−54%). Figure 10c reveals a linear dependency of the stress at break with the PCM content, whereas HGMs do not influence it. When σ_B_ is normalized with density, as represented in Figure 10d, its trend is similar to that shown by the elastic modulus, thus leading to similar conclusions.

One important peculiarity of syntactic foams, which allows their wide application in the marine environment [46,47], is their high compression strength. Therefore, quasi-static compression tests were conducted on the prepared foams, to study the trends of the compressive elastic modulus (E_C_), the specific elastic modulus (E_C_/ρ), the stress at 20% of strain (σ_20_), and the specific stress at 20% of strain (σ_20_/ρ) with the amount of PCM and HGMs. Figure 11 shows representative stress–strain curves of seven foams obtained during the compression tests, while Table 6 and Figure 12a–d show these results numerically and graphically.

In accordance with tensile test results, Figure 11 shows that the compressive performances increase upon the addition of HGMs, whereas they decrease with introduction of PCMs. As reported in Table 6, the elastic modulus increases from 685 MPa of the neat PP matrix up to 1011 MPa of the H40-P0 sample (+49%) and decreases down to 421 MPa for the H0-P30 sample (−39%). Furthermore, the stress–strain curves of the compositions containing both HGM and PCM (green curves in Figure 11) lie in the middle, with lower values of elastic modulus than the matrix (−15% for H20-P20). Figure 12a better represents the trends of the compressive modulus, which are similar to those observed in tensile mode. Similar to what was observed in the tensile tests, the elastic modulus (Figure 12a) shows a proportionality with the degree of substitution of PCMs with HGMs (moving horizontally from right to left). This trend is even more evident in Figure 12b, where the specific compressive modulus is shown. Considering the results of stress at 20% of strain, reported in Table 6, it is clear that introduction of HGMs leads to a retention of the σ_20_ values, while addition of PCMs decreases it. This trend is even better represented in Figure 12c. The σ_20_ decreases from 54.3 MPa of the matrix down to 27.5 MPa for the H0-P30 foams (−49%), while it remains stable with addition of HGMs. This trend becomes much more similar to that of the compressive elastic modulus when specific σ_20_ values are considered (Figure 12d). In conclusion, these results confirm the positive compressive properties of the traditional syntactic foams (i.e., those containing only HGMs), while incorporation of PCMs leads to a drop in compressive performances. Good adhesion between HGMs and the matrix plays an important role in increasing mechanical performances, especially when normalized properties are considered.

The stress intensity factor (K_IC_) describes the resistance offered by materials to crack propagation, and it is a fundamental property in structural design. Due to the ductile nature of PP, it is very difficult to measure this property under quasi-static conditions, following the ASTM D5045 standard. For this reason, these tests were conducted under Charpy impact conditions at a testing speed of 1.5 m/s, as suggested by the ISO 13586 standard. The results of these tests are reported in Table 7, while the ternary diagram with the K_IC_ values is shown in Figure 13.

These results clearly show that K_IC_ values are mainly influenced by the content of PCMs. In fact, incorporation of HGMs does not strongly affect the fracture toughness of the foams, whereas addition of PCMs reduces K_IC_ from 2.12 MPa∙m^1/2^ of the PP matrix down to 1.24 MPa∙m^1/2^ of the H0-P30 sample (−41%). In compositions mainly containing HGMs, the correlation between K_IC_ and the PCM content is even more evident. This behavior can be associated with the poor mechanical performances of PCMs and their poor adhesion to the PP matrix. On the contrary, HGMs are capable of retaining the pristine fracture toughness of the PP, and also reducing the overall weight of the final composite.

### 2.4. General Comparison of the Physical Properties of the Prepared Syntactic Foams

To perform a general comparison between the most important physical properties of the prepared foams, Figure 14 reports in a single radar plot a direct comparison of the performances of the six most representative compositions. This comparative analysis has been conducted in terms of specific tensile modulus (E_t_/ρ), specific tensile stress at break (σ_B_/ρ), specific compressive elastic modulus (E_C_/ρ), specific compressive stress at 20% of strain (σ_20_/ρ), stress intensity factor (K_IC_), specific volume (ν, that is 1/ρ), melting enthalpy (ΔH_m_), and thermal resistance at 45 °C (R_λ_). 

Thanks to the color code of Figure 14, it is possible to differentiate between three families of composites, i.e., samples containing only HGMs (blue), only PCMs (red), and containing both (green), plus the neat matrix (black). This graph shows the clear dominance in mechanical properties, lightness, and thermal resistivity of compositions containing only HGMs (blue). On the flip side, as expected, they lack in TES capability. On the contrary, compositions containing only PCMs (red) show the best TES properties but, unfortunately, the same cannot be said for the mechanical properties. A good compromise can be achieved with mixed compositions containing both fillers (green). For these samples, the mechanical properties are improved by the presence of HGMs, while TES properties are still interesting for many applications, making these compositions very interesting for their multifunctionality. In conclusion, if mechanical performances are the main goal and TES properties are not required, H40-P0 could be the best choice because it shows interesting mechanical performances, especially when the components’ weight should be minimized. If TES properties are the main goal and the material is not subjected to severe mechanical stresses, H0-P30 could be the best choice. By combining the two fillers it is possible to develop very different types of composites that could fulfill various needs, thus creating versatile materials that can be tuned for a specific application. 

## 3. Materials and Methods

### 3.1. Materials

The polypropylene used for this work, Moplen HP456J, was a commercial grade of PP commonly used for sheet molding operations, and it was kindly provided by Lyondell Basell (U.S.). Table 8 reports the most important thermo-mechanical properties of the utilized PP.

The PCM used in this work (MPCM57D) was an encapsulated paraffin produced by Microtek laboratories Inc. (Dayton, OH, USA), having a melting temperature of about 57 °C and a melting enthalpy of 210 J/g. The paraffinic core is enclosed in a melamine-formaldehyde capsule thermally stable until 250 °C. The capsules’ mean diameter (D50) is close to 34 μm, while their density is 0.95 g/cm^3^. 3 M Italia Srl (Pioltello, Italy) provided the hollow glass microspheres iM16K HGMs used in this work. They are made of soda-lime-borosilicate glass, have a density of 0.45 g/cm^3^_,_ and possess a crush strength (90% survival) near 110 MPa, thanks to their high wall thickness/diameter ratio. They possess a mean particle size (D_50_) equal to 27 μm, and a thermal conductivity of 0.153 W/(m∙K). The HGMs were silanized with γ-aminopropyl-triethoxysilane (APTES) before use. The compatibilizer used in this work was a maleic anhydride grafted polypropylene (MA-g-PP), commercially known as Copoline CO/PP H60, provided by Auserpolimeri Srl (Italy). The compatibilizer density is 0.87 g/cm^3^, while the melt flow index (MFI) at 230 °C and 2.16 kg is 180 g/10 min. All the materials were used as received.

### 3.2. Sample Preparation

#### 3.2.1. Silanization of HGMs

The chemical process of silanization aims to improve the adhesion between two incompatible surfaces by joining them with particular molecules known as silanes [43]. This process is typically conducted on glass fibers to improve their adhesion to the matrix. For the same reason, this process was used on HGMs and the main procedure steps are listed as follows. This procedure is a result of an optimization performed based on the literature information [48,49,50,51,52].
➢Hydroxylation
▪12 g of HGMs were added in a 0.5 M solution of NaOH▪The solution was stirred using a magnetic anchor at 90 °C for 1 h▪After the solution reached room temperature it was filtered and washed with distilled water until a pH = 7 was reached▪The filter with the washed HGMs was then dried in a vacuum oven at 80 °C for 12 h➢Silanization
▪10 g of HGMs were mixed with a preheated solution (200 mL, 70 °C) of 50/50 mass of ethanol and distilled water plus 2 g of APTES.▪After the mixture was mixed for 1.5 h the solution was cooled and filtered again as in the hydroxylation step▪The silanized HGMs were filtered and then dried in a vacuum oven for 12 h at 80 °C.

#### 3.2.2. Preparation of Multifunctional SFs

The constituents (PP, compatibilizer, silanized HGMs, and PCM) were mixed at different relative amounts by using a Thermo Haake Rheomix 600 internal mixer equipped with counter-rotating rotors, operating at 30 rpm and 200 °C for 5 min. The obtained mixtures were placed in aluminum plates and hot pressed at 210 °C, applying a pressure of 545 kPa for 5 min. Subsequently, the obtained syntactic foam plates were milled to prepare specimens with different shapes, to be utilized for the different characterization techniques. These specimens were prepared by using molds with specific dimensions, at a hot-pressing temperature of 200 °C, and applying a pressure of 4 bar. 

To characterize this new ternary system (PP/HGM/PCM), ten compositions were analyzed as represented in Figure 15 and Table 9. In Figure 15 PP remains as the matrix, composed of the polypropylene with the compatibilizer. The concentration of compatibilizer (i.e., 5% by volume of the matrix) was optimized through preliminary tests.

### 3.3. Experimental Techniques

#### 3.3.1. Rheological and Morphological Analysis

A simple way to evaluate the processing window of thermoplastic materials is to perform rheological analyses, monitoring the trends of the storage and loss modulus (G′ and G″ respectively), and the viscosity as a function of the shear rate. In this work, a DHR-2 rheometer (TA instrument, New Castle, DE, USA), operating in plate–plate geometry (25 mm in diameter, 1 mm gap) at 200 °C was utilized, both in frequency sweep and flow sweep configurations. The first testing mode was used to determine G′ and G″ trends in a frequency interval from 0.1 to 600 rad/s with a fixed strain amplitude of 1%, while the second testing configuration was used to measure the viscosity (η) at values of shear rate ($\dot{\gamma }$) between 0.01 to 200 1/s. The frequency at which G′ crosses G″ is defined as the crossover frequency *(*$\omega_{c}$), which represents the threshold between the viscous and elastic behavior regions. When G″ > G′, the viscous response dominates the elastic one. For these tests, only one specimen was considered for each composition.

The density (ρ_exp_) of the prepared samples was determined through a Micromeritics AccuPyc 1330TC (Micromeritics Instrument Corp., Norcross, GA, USA) helium pycnometer, equipped with a testing cell of 1 cm^3^ and operating at a temperature of 23 °C. The theoretical density of the selected compositions was calculated by using Equation (1), while the void content of the tested samples was evaluated through Equation (2).
(1)$$\rho_{\mathrm{th}}=\overset{n}{\underset{i=1}{\sum }}\rho_{i}\cdot \phi_{i}$$
where ρ_i_ is the density of the i-th component and ϕ_i_ is its volume fraction.
(2)$$\mathrm{void}\;\mathrm{content}\%=\frac{\rho_{\mathrm{th}}-\rho_{\exp}}{\rho_{\mathrm{th}}}$$

Scanning electron microscopy (SEM) was carried out on the Pt-Pd sputtered cryo-fractured sample surfaces, to evaluate the microstructure of the different compositions. For this aim, a Zeiss Supra 40 microscope (Carl Zeiss AG, Oberkochen, Germany), operating at an acceleration voltage of 3.5 kV, was utilized.

#### 3.3.2. Evaluation of the Thermal Properties

TES capability is the multifunctional property intended to be added to these syntactic foams through incorporation of PCMs. For this aim, a Mettler DSC30 instrument (Mettler Toledo LLC, Columbus, OH, USA) was used, testing specimens of 10 mg inserted in 40 μL aluminum crucibles under a nitrogen flow of 10 mL/min. A total of three scans between −50 and 220 °C at a heating/cooling rate of 10 °C/min were carried out. In this way, it was possible to measure the temperature of fusion and crystallization (T_m_ and T_c_) and the corresponding enthalpy values ($∆H_{m}$ and $∆H_{c}$) of the constituents and the prepared foams. The specific heat capacity (c_p_) of the foams was evaluated through these tests, by following the procedure described in the ASTM E1269 standard. Three specimens were tested for each composition. 

Moreover, the TES capability of these foams was also evaluated by monitoring the evolution of their temperature upon heating/cooling stages. Therefore, a type-K thermocouple connected to a recording system was inserted in a pre-machined blind hole (2 mm in diameter and 17.5 mm in depth) performed on a cubic specimen (35 mm in size). A total of seven compositions were considered for this test. After inserting the thermocouple, the cubes were placed in an oven preheated to 100 °C, and the time required to increase their temperature from 35 to 80 °C (t_35–80 °C_) was recorded. After that, a temperature of about 95 °C was reached, the cubes were extracted from the oven to cool, and the cooling time of the samples from 80 to 35 °C (t_80–35 °C_) was determined.

A simple and fast way to measure thermal diffusivity ($\alpha \left(T\right)$) and thermal conductivity ($\lambda \left(T\right)$) is through a light flash analyzer (LFA). An LFA 467 machine (Netzsch Holding, Selb, Germany) was used to test disc specimens having a diameter of 12.5 mm and a thickness of 1 mm, pre-coated with graphite spray. These tests were performed at three different temperatures, 20, 45, and 70 °C. By knowing the specific heat capacity of the samples from DSC measurements (c_p_(T)) and their density ($\rho \left(T\right)$) from pycnometric measurements, the thermal conductivity was then calculated as reported in Equation (3).
(3)$$\lambda \left(T\right)=\alpha \left(T\right)\cdot \rho \left(T\right)\cdot c_{p}\left(T\right)$$

#### 3.3.3. Evaluation of the Mechanical Properties of SFs

Quasi-static tensile and compression tests were performed on the prepared foams, and their fracture toughness (K_IC_) under impact conditions was also evaluated. All these analyses were performed at 25 °C and 50% relative humidity. 

An Instron 5969 universal testing machine (Instron, Turin, Italy), equipped with a 1 kN load cell, was used to perform tensile tests following the ISO 5278-2 standard. Specimens of 1BA having a gage length of 55 mm, were utilized. The determination of the elastic modulus (E_t_) was performed at a testing speed of 0.25 mm/min, by using an extensometer having a gauge length of 25 mm and the stress levels associated with deformations of 0.05% and 0.25% were considered. Tensile properties at break were determined at a crosshead speed of 20 mm/min without the extensometer until failure was reached. In this way, it was possible to determine the stress and the strain at yield (σ_y_, ε_y_) and at break (σ_B_, ε_B_). At least ten specimens were tested for each composition.

By using the same testing machine, equipped with a 10 kN load cell, it was possible to study the quasi-static compression properties, following the ASTM D695 standard. A total of nine cubic specimens (1 × 1 × 1 cm^3^) were tested for each composition at a crosshead speed of 1.3 mm/min. In this way, it was possible to define the elastic modulus at compression (E_C_), by using the tangent method, and the stress at 20% of strain (σ_20_).

Due to the tough nature of polypropylene, the determination of the fracture toughness (K_IC_) of the prepared foams was performed under impact conditions, following ISO 13586 and ISO 17281 standards. A Ceast 3549/000 impact testing machine (Instron, Turin, Italy) equipped with a 5 J instrumented hammer was used. A three-point bending configuration, with a span length of 62 mm was adopted. A starting impact angle of 39° was set, to hit the specimens with a speed of 1 m/s. The acquisition system was capable of measuring for 8 ms after the impact at a sampling rate of 2 µs. For each composition, ten rectangular specimens (10 × 80 × 4 mm^3^) with a sharp notch having a depth of 5 mm were tested. The procedure for calculating K_IC_, extensively described in ISO 13586 and ASTM D5045 standards, was not reported for the sake of brevity.

### 3.4. Statistical Analysis of the Experimental Data

As reported in our previous articles [8,42], due to the system’s complexity the analysis of the properties could be quite difficult without the application of a rigorous statistical approach. By using RStudio v.1.4.1103 software (RStudio, Inc., Boston, MA, USA) it was possible to fit the experimental results of the samples with different compositions with a quadratic linear model called the “Scheffé quadratic model” by calling “lm” and “ModelPlot” functions in the software (more details can be found in [42]). Those functions were used mainly to plot the ternary phase diagrams and to report the most important statistical data like the goodness of fit (R^2^_adj_) and the residual standard error of the model (S_y.x_) defined by Equations (4) and (5). The average coefficient of variance (ACV) was also considered, and its expression is reported in Equation (6).
(4)$$R_{\mathrm{adj}}^{2}=1-\left[\frac{\left(1-R^{2}\right)\cdot \left(n-1\right)}{n-k-1}\right]$$
(5)$$S_{x.y}=\sqrt{\frac{\sum \left({\mathrm{residual}}^{2}\right)}{n-k}}$$
where n are the experimental values and k is the number of parameters fit by regression, and R^2^ is the coefficient of determination that gives information about the goodness of fit of a model.
(6)$$\mathrm{ACV}=\frac{1}{N}\overset{N}{\underset{J=1}{\sum }}\left(100\cdot \frac{t_{J}}{{\bar{x}}_{J}}\cdot \sqrt{\frac{\sum_{i=1}^{n_{J}}{\left(x_{\mathrm{Ji}}-{\bar{x}}_{J}\right)}^{2}}{n_{J}\cdot \left(n_{J}-1\right)}}\right)$$
where N is the number of considered compositions, t_J_ is the t-value extrapolated from the t-student distribution for the j-th composition, ${\bar{x}}_{J}$ is the average value of the j-th composition, $n_{J}$ is the number of specimens of each composition, and $x_{\mathrm{Ji}}$ is the experimental value of the i-th tested specimen of the j-th composition.

## 4. Conclusions

In this paper, HGMs and PCMs were incorporated at different relative amounts into a compatibilized polypropylene matrix to develop novel multifunctional syntactic foams with thermal energy storage capability. Rheological tests showed that HGMs increased the viscosity of the molten material at 200 °C while PCMs slightly reduced it, as a consequence of the PCMs breakage during the production step. SEM micrographs confirmed the partial breakage of the PCM microcapsules and highlighted a good adhesion between the matrix and HGMs, thanks to the silanization process performed on the surface of the glass beads and the presence of the compatibilizer. As expected, the density of these syntactic foams was diminished by addition of HGMs, but also the introduction of PCMs led to a density drop, probably because of the increased porosity generated by the partial evaporation of the PCM that leaked out from the broken capsules. 

Both DSC and temperature profile analyses confirmed the PCM’s capability to impart TES properties to the foams, with the H0-P30 sample capable of storing 57 J/g in the first DSC heating scan. Both HGMs and PCMs diminished the thermal conductivity of the foams, which was reduced from 0.268 W/m∙K for the PP matrix down to 0.190 W/m∙K for H40-P0 samples (−30%) and 0.183 W/m∙K for H0-P30 foam (−33%). Compositions containing both fillers showed a further thermal conductivity reduction, up to 0.136 W/m∙K for the H27-P7 sample (−50%).

The compositions containing only HGMs proved to be the best choice from a mechanical point of view. Quasi-static tensile tests reported an increment in the mechanical performances (especially of the elastic modulus) upon introduction of HGMs, while PCMs tended to diminish them. A similar trend was found in compression, where addition of HGMs significantly increased the elastic modulus and the compressive strength. K_IC_ was not affected by the presence of HGM, while it was impaired by the presence of PCM, due to their poor strength and the limited interfacial addition with the PP matrix. 

It can be therefore concluded, that incorporation of PCMs into these PP syntactic foams resulted in a material with a set of properties that could be properly tuned, depending on the final purpose, by simply acting on the filler amount. This aspect opens up new opportunities for these systems in applications where TES properties are strongly required, such as in electronics, automotive, refrigeration, and aerospace. In the future, by acting on the production process, the PCMs failure rate could be significantly reduced, thus improving the final performances of these materials.

## Figures and Tables

**Figure 1 molecules-27-08520-f001:**
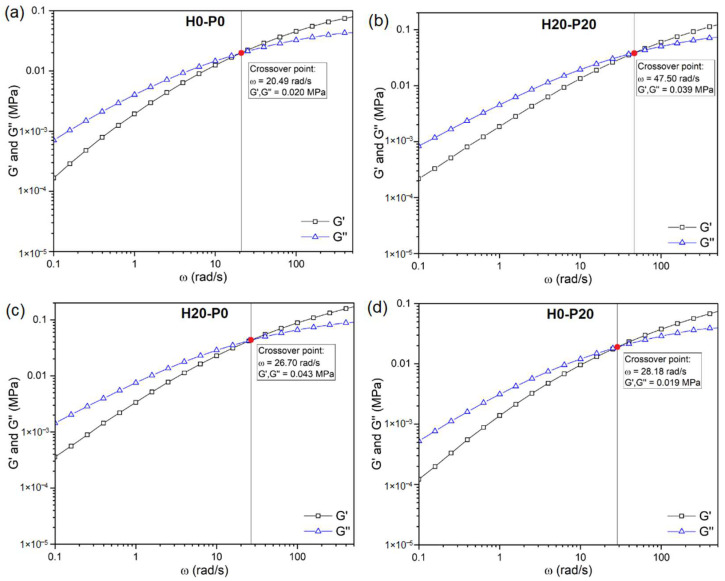
Results of frequency sweep tests on different compositions with the indication of the crossover point. (**a**) H0-P0, (**b**) H20-P20, (**c**) H20-P0, and (**d**) H0-P20 samples.

**Figure 2 molecules-27-08520-f002:**
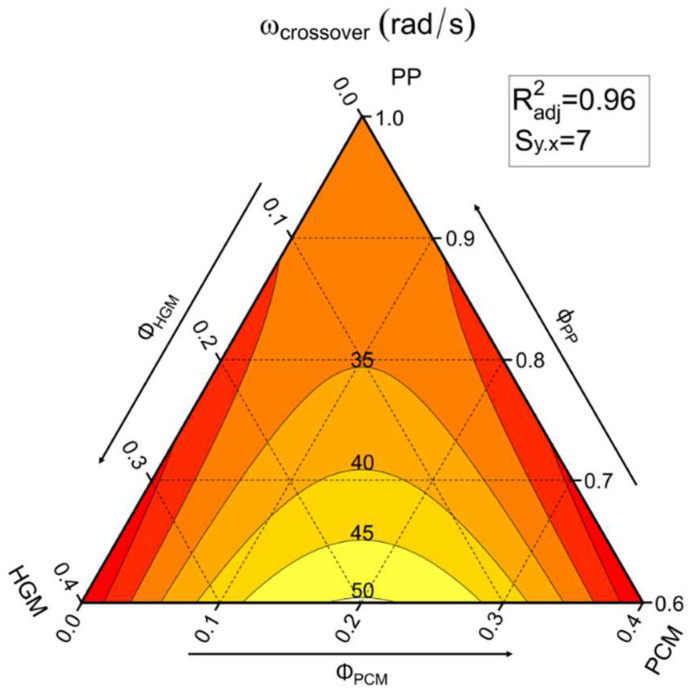
Ternary diagram representing the frequency values of the crossover point derived from rheological tests on the prepared foams.

**Figure 3 molecules-27-08520-f003:**
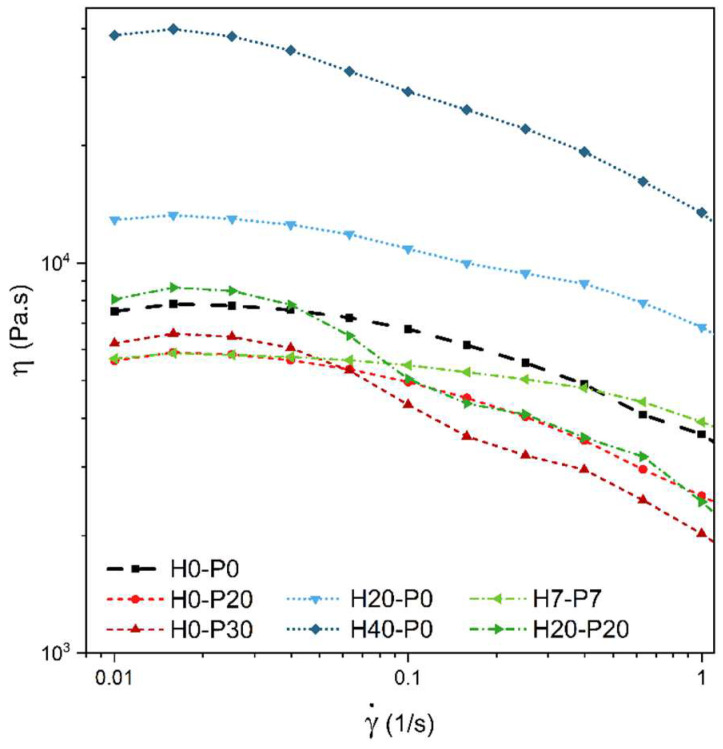
Viscosity (η) as a function of the shear rate ($\dot{\gamma }$) from flow sweep rheological tests on the prepared foams.

**Figure 4 molecules-27-08520-f004:**
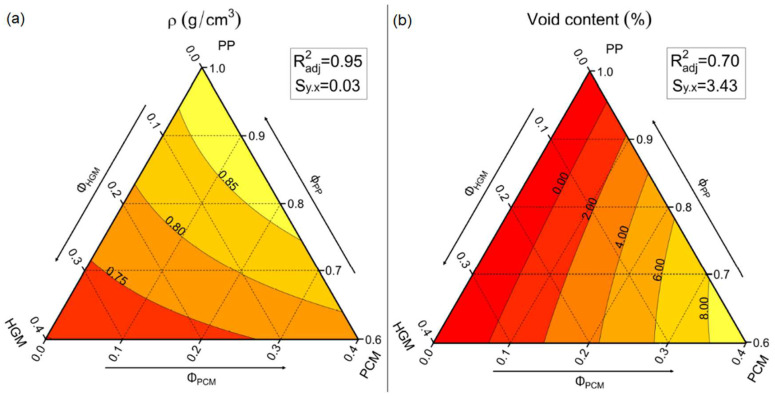
Ternary diagram of (**a**) the experimental density and (**b**) the void content of the prepared foams.

**Figure 5 molecules-27-08520-f005:**
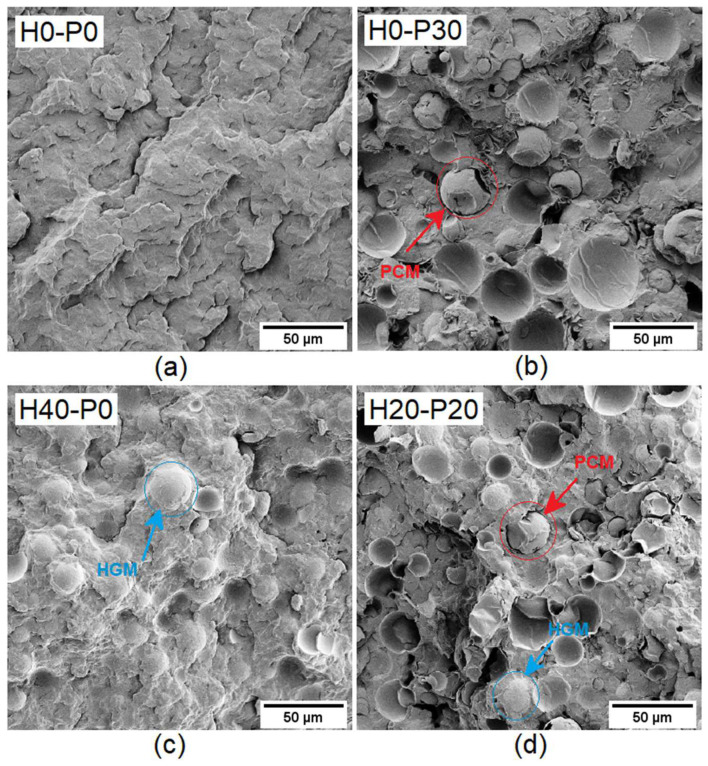
SEM micrographs of the cryo-fractured surfaces of four selected compositions. (**a**) H0-P0, (**b**) H0-P30, (**c**) H40-P0, and (**d**) H20-P20.

**Figure 6 molecules-27-08520-f006:**
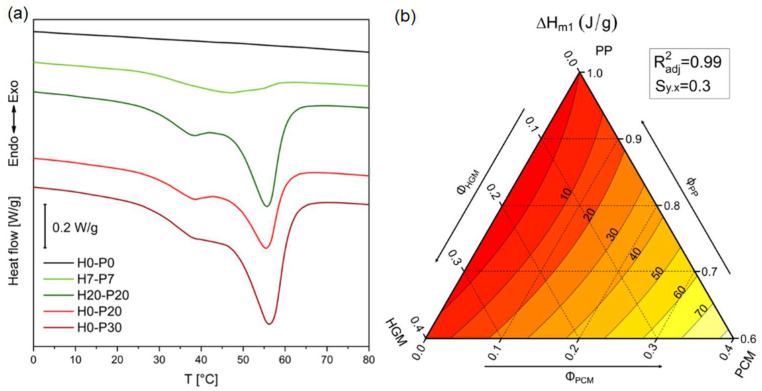
(**a**) Representative DSC thermograms of the prepared foams (first heating scan), (**b**) ternary diagram of the melting enthalpy values (first heating scan).

**Figure 7 molecules-27-08520-f007:**
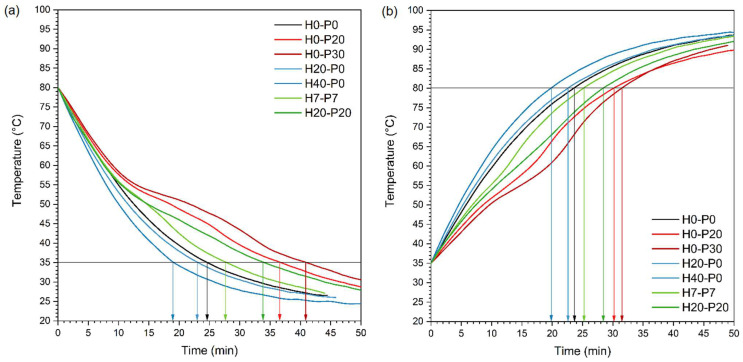
Temperature profiles of the analyzed samples in (**a**) cooling and (**b**) heating stages.

**Figure 8 molecules-27-08520-f008:**
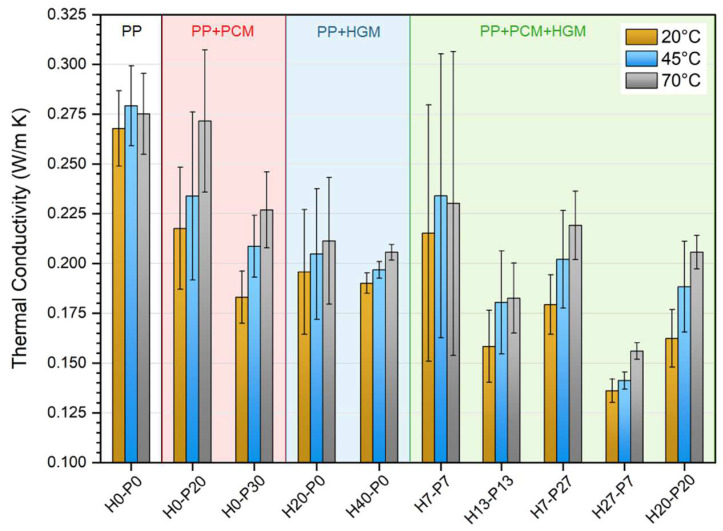
Thermal conductivity values of the prepared foams from LFA analysis.

**Figure 9 molecules-27-08520-f009:**
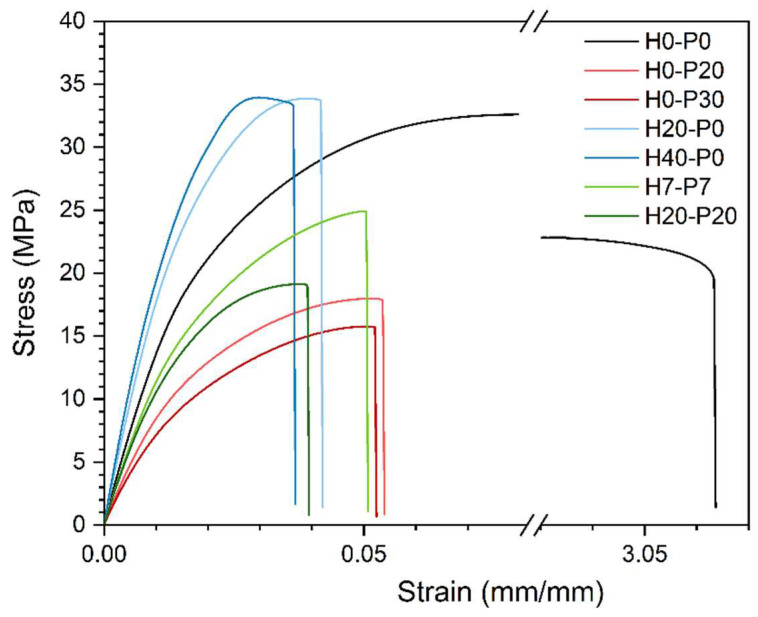
Tensile stress–strain curves for some representative compositions.

**Figure 10 molecules-27-08520-f010:**
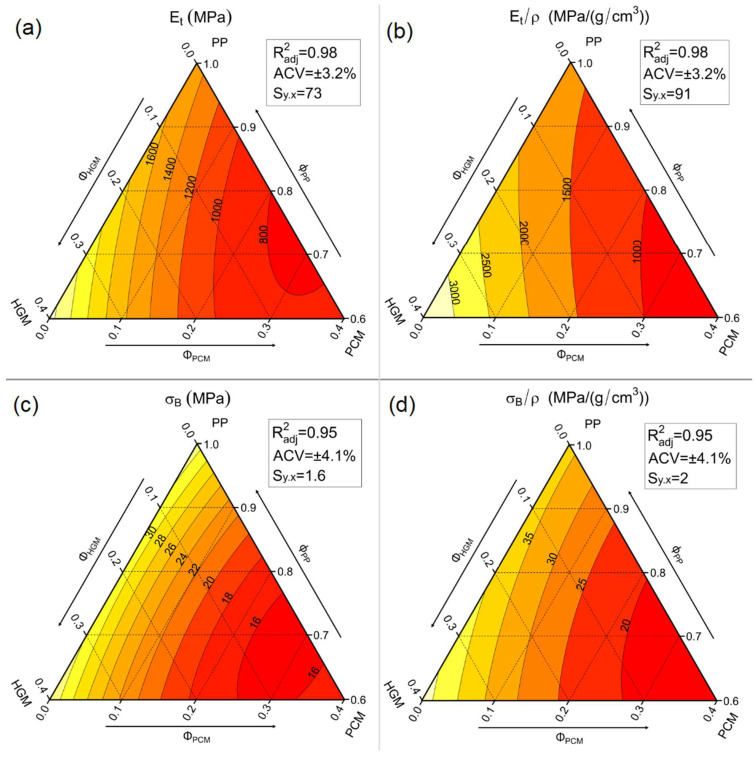
Ternary diagrams of the quasi-static tensile properties of the considered syntactic foams. (**a**) Elastic modulus (E_t_), (**b**) specific elastic modulus (Et/ρ), (**c**) tensile stress at break (σ_B_), and (**d**) specific stress at break (σ_B_/ρ).

**Figure 11 molecules-27-08520-f011:**
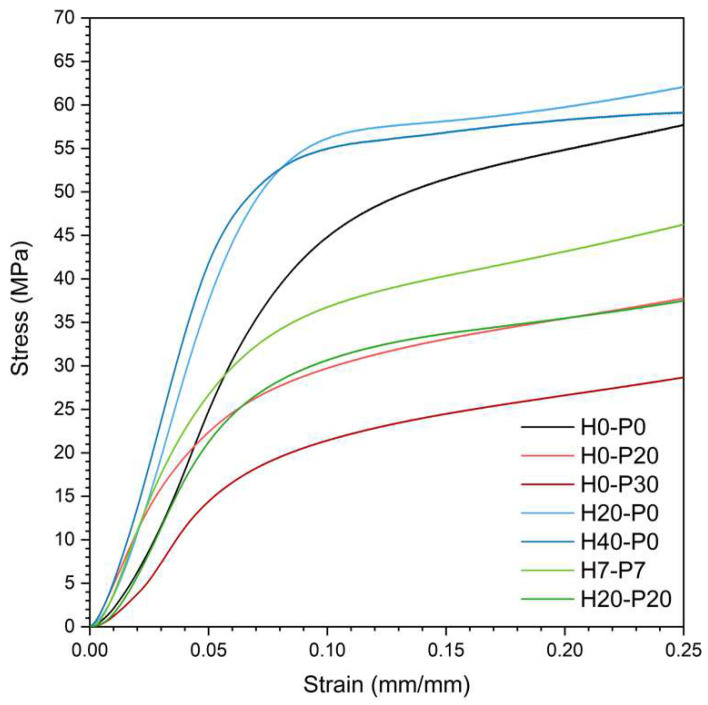
Compressive stress–strain curves for some selected representative compositions.

**Figure 12 molecules-27-08520-f012:**
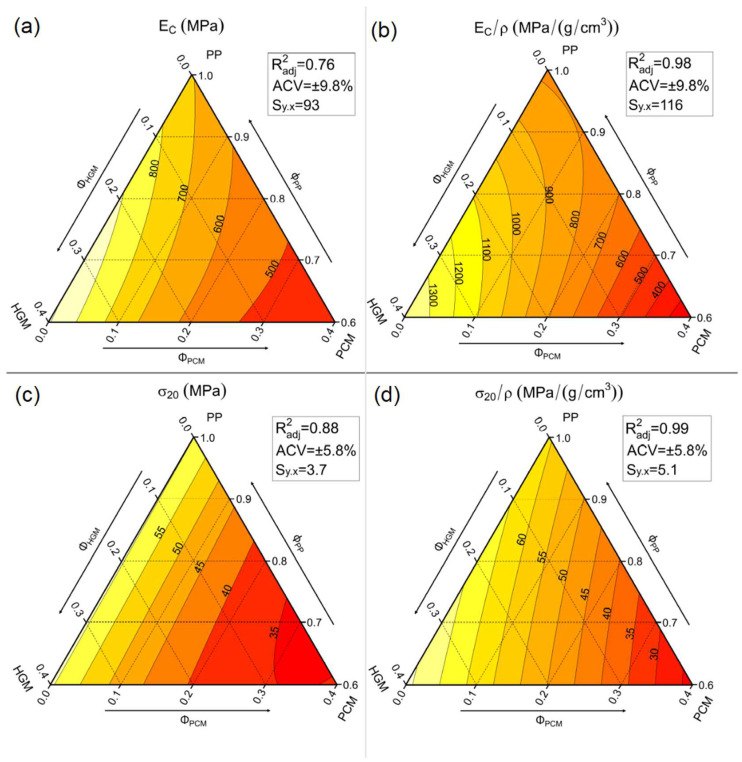
Ternary diagrams of the quasi-static compressive properties of the prepared syntactic foams. (**a**) Compressive modulus (E_C_), (**b**) specific compressive modulus (E_C_/ρ), (**c**) compressive stress at 20% (σ_20_), and (**d**) specific compressive stress at 20% (σ_20_/ρ).

**Figure 13 molecules-27-08520-f013:**
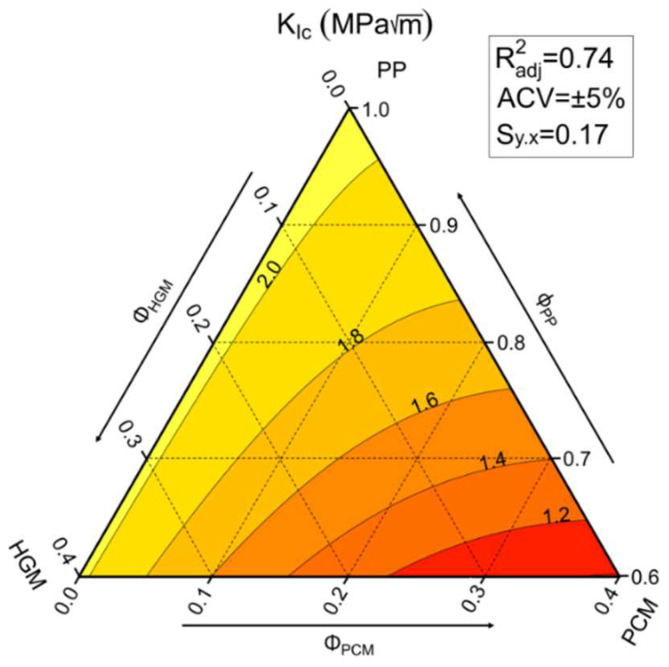
Ternary diagram of the critical stress intensity factor (K_IC_) under impact conditions of the prepared foams.

**Figure 14 molecules-27-08520-f014:**
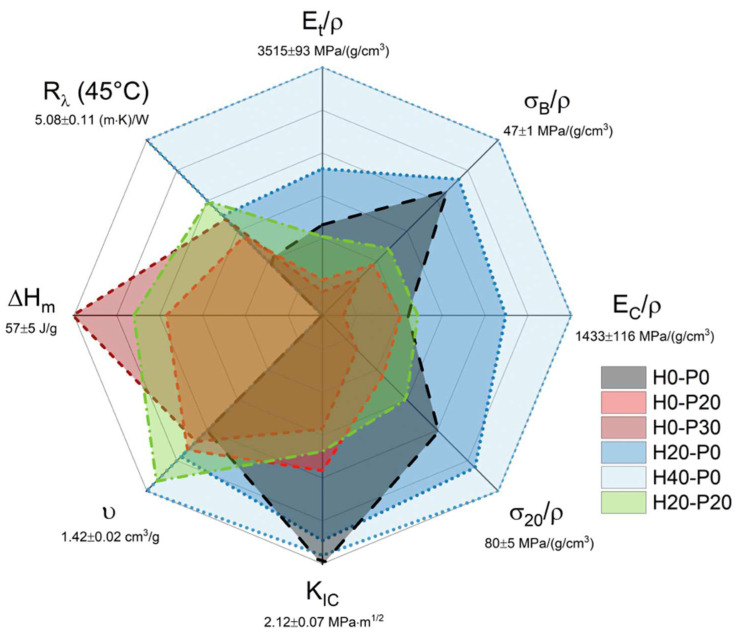
Radar diagram showing a graphical comparison of the physical properties of the prepared foams.

**Figure 15 molecules-27-08520-f015:**
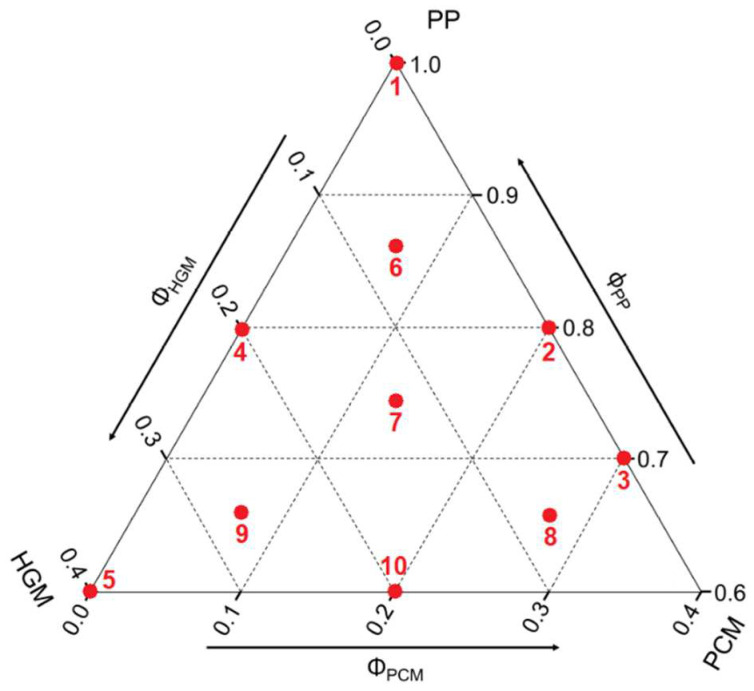
Representation of the selected compositions (highlighted by the red dots) on the ternary diagram. The numbers (1–10) on each dot correspond to the sample numbers indicated in Table 9.

**Table 1 molecules-27-08520-t001:** Crossover point coordinates of all the analyzed samples.

Sample	Crossover Point		Constituents
	ω (rad/s)	G′, G″ (MPa)	
H0-P0	20.5	0.020	PP
H0-P20	28.2	0.019	PP + PCM
H0-P30	35.0	0.024	
H20-P0	26.7	0.043	PP + HGM
H40-P0	21.6	0.098	
H7-P7	36.0	0.027	PP + HGM + PCM
H13-P13	34.4	0.034	
H7-P27	39.2	0.023	
H27-P7	42.7	0.031	
H20-P20	47.5	0.039	

**Table 2 molecules-27-08520-t002:** Experimental and theoretical density, and void content of the prepared foams.

Sample	$\rho_{\exp}$ (g/cm^3^)	$\rho_{\mathrm{th}}$ (g/cm^3^)	Void Content (%)
H0-P0	0.90	0.90	0.0
H0-P20	0.83	0.90	8.1
H0-P30	0.86	0.91	5.0
H20-P0	0.81	0.81	0.0
H40-P0	0.70	0.72	1.8
H7-P7	0.80	0.87	7.4
H13-P13	0.77	0.84	9.1
H7-P27	0.83	0.89	5.8
H27-P7	0.74	0.78	5.0
H20-P20	0.73	0.82	10.2

**Table 3 molecules-27-08520-t003:** DSC results of the first heating and cooling scans of the prepared foams. The final column shows the theoretical melting enthalpy (ΔH_m,th_), referred to the nominal PCM content in the foams.

Sample	T_m1_ (°C)	ΔH_m1_ (J/g)	T_c_ (°C)	ΔH_c_ (J/g)	ΔH_m,th_ (J/g)
H0-P0	-	0.0	-	0.0	0.0
H0-P20	55.6	36.5	37.2	36.5	41.4
H0-P30	55.7	58.4	37.3	58.2	62.1
H20-P0	-	0.0	-	0.0	0.0
H40-P0	-	0.0	-	0.0	0.0
H7-P7	46.9	8.2	35.1	8.2	13.7
H13-P13	54.6	22.4	36.6	22.3	27.5
H7-P27	55.3	53.0	37.6	51.5	57.3
H27-P7	54.6	11.4	36.4	11.3	13.7
H20-P20	55.2	43.9	37.5	43.4	41.4

**Table 4 molecules-27-08520-t004:** Results of thermal profiling tests on the prepared foams.

Sample	t_35–80_ (min)	t_80–35_ (min)
H0-P0	23.5	24.6
H0-P20	30.1	36.7
H0-P30	31.4	41.0
H20-P0	22.6	23.1
H40-P0	19.8	19.1
H7-P7	25.1	27.8
H20-P20	28.4	34.0

**Table 5 molecules-27-08520-t005:** Tensile mechanical properties of all the considered compositions.

Samples	E_t_ (MPa)	$E_{t}/\rho$ (MPa/(g/cm^3^))	σ_B_ (MPa)	$\sigma_{B}/\rho$ (MPa/(g/cm^3^))	ε_B_ (mm/mm)
H0-P0	1509 ± 19	1677 ± 21	33.5 ± 0.5	37.2 ± 0.5	2.320 ± 1.970
H0-P20	845 ± 55	1018 ± 66	17.4 ± 0.8	21.0 ± 1.0	0.048 ± 0.004
H0-P30	758 ± 15	881 ± 17	15.5 ± 0.4	18.0 ± 0.5	0.048 ± 0.003
H20-P0	1873 ± 61	2312 ± 75	31.8 ± 1.3	39.3 ± 1.6	0.042 ± 0.007
H40-P0	2480 ± 66	3543 ± 94	33.8 ± 0.8	48.3 ± 1.1	0.037 ± 0.006
H7-P7	1183 ± 65	1479 ± 81	23.6 ± 2.3	29.5 ± 2.9	0.044 ± 0.010
H13-P13	1159 ± 31	1505 ± 39	20.9 ± 0.5	27.1 ± 0.7	0.041 ± 0.002
H7-P27	751 ± 15	905 ± 18	14.6 ± 0.8	17.6 ± 1.0	0.038 ± 0.003
H27-P7	1649 ± 37	2228 ± 50	24.5 ± 0.9	33.1 ± 1.2	0.030 ± 0.003
H20-P20	1117 ± 39	1530 ± 53	17.9 ± 0.8	24.5 ± 1.1	0.035 ± 0.003

**Table 6 molecules-27-08520-t006:** Compressive properties of the prepared foams.

Samples	E_C_ (MPa)	$E_{C}/\rho$ (MPa/(g/cm^3^))	σ_20_ (MPa)	$\sigma_{20}/\rho$ (MPa/(g/cm^3^))
H0-P0	685 ± 120	761 ± 133	54.3 ± 3.3	60.3 ± 3.7
H0-P20	602 ± 40	725 ± 48	33.9 ± 1.9	40.8 ± 2.3
H0-P30	421 ± 39	490 ± 45	27.5 ± 1.3	32.0 ± 1.5
H20-P0	937 ± 116	1157 ± 143	58.4 ± 3.0	72.1 ± 3.7
H40-P0	1011 ± 82	1444 ± 117	56.7 ± 3.6	81.0 ± 5.1
H7-P7	714 ± 43	893 ± 54	45.3 ± 2.5	56.6 ± 3.1
H13-P13	617 ± 65	801 ± 84	39.8 ± 1.9	51.7 ± 2.5
H7-P27	543 ± 55	654 ± 66	31.2 ± 1.4	37.6 ± 1.7
H27-P7	793 ± 83	1072 ± 112	41.4 ± 3.6	56.0 ± 4.9
H20-P20	582 ± 42	797 ± 58	35.7 ± 2.4	48.9 ± 3.3

**Table 7 molecules-27-08520-t007:** K_IC_ results of the prepared foams.

Samples	K_IC_ (MPa∙m^1/2^)
H0-P0	2.12 ± 0.07
H0-P20	1.51 ± 0.06
H0-P30	1.24 ± 0.06
H20-P0	1.96 ± 0.06
H40-P0	2.06 ± 0.14
H7-P7	1.88 ± 0.14
H13-P13	1.61 ± 0.09
H7-P27	1.25 ± 0.05
H27-P7	1.65 ± 0.10
H20-P20	1.38 ± 0.07

**Table 8 molecules-27-08520-t008:** Main thermo-mechanical properties of Moplen HP456J from the technical datasheet.

Property	Value	Unit	Test Method
Melt flow rate	3.4	g/10 min	ISO 1133-1
Density	0.90	g/cm^3^	ISO 1183-1
Flexural modulus	1500	MPa	ISO 178
Tensile stress at yield	34	MPa	ISO 527-1-2
Tensile stress at break	21	MPa	ISO 527-1-2
Tensile strain at break	200	%	ISO 527-1-2
Vicat (A50)	156	°C	ISO 306
HDT (0.45 MPa)	91	°C	ISO 75B-1-2

**Table 9 molecules-27-08520-t009:** List of the prepared samples with their nominal composition.

#	Compositions	Compatibilizer (vol%)	PP (vol%)	HGM (vol%)	PCM (vol%)
1	H0-P0	5.00	95.00	0.00	0.00
2	H0-P20	4.00	76.00	0.00	20.00
3	H0-P30	3.50	66.50	0.00	30.00
4	H20-P0	4.00	76.00	20.00	0.00
5	H40-P0	3.00	57.00	40.00	0.00
6	H7-P7	4.33	82.33	6.70	6.70
7	H13-P13	3.67	69.67	13.33	13.33
8	H7-P27	3.33	63.33	6.67	26.67
9	H27-P7	3.33	63.33	26.67	6.67
10	H20-P20	3.00	57.00	20.00	20.00

## Data Availability

Not applicable.

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
