# Peer review of "Development of Novel Polypropylene Syntactic Foams Containing Paraffin Microcapsules for Thermal Energy Storage Applications"

_molecules, 2022, doi:10.3390/molecules27238520_

Round 1
Reviewer 1 Report
In this work, HGMs and PCMs were incorporated at different relative amounts into a compatibilized PP matrix to develop novel multifunctional syntactic foams with thermal energy storage capability. The results revealed that incorporating PCMs into these PP syntactic foams resulted in a material with a set of properties that could be properly tuned, depending on the final purpose, by simply acting on the filler amount.
Some detailed comments:
1. The writing of the introduction needs to be more concerned with specific issues, especially the progress in the relevant areas of the study. Reduce the statements about basic knowledge.
2. SEM surface sample was obtained by cryo-fracturing,so can this result confirm the PCM capsule partial breakage “during the compounding operations”?
3. In Table 8, which property belongs to “the most important properties of the utilized PP”?
Author Response
Reviewer 1
- The writing of the introduction needs to be more concerned with specific issues, especially the progress in the relevant areas of the study. Reduce the statements about basic knowledge.
The authors thank the reviewer for the comment. The introduction has been modified as suggested by the reviewer, shortening the text in some general parts.
- SEM surface sample was obtained by cryo-fracturing, so can this result confirm the PCM capsule partial breakage “during the compounding operations”?
The authors thank the reviewer for the comment. The SEM pictures of this work show capsules damaged differently with respect to what has been found in our previous work with epoxy resin [1]. In that case, the cryo-fracturing process clearly led to the breakage of the capsules, leaving in most cases the solid PCM inside. In the present case, SEM micrographs show a different situation, in which voids capsules are present, while other capsules are only slightly damaged. It can therefore thought that these damages were produced only by the shear stresses applied during the compounding process, and not by the cryofracturing operations.
- In Table 8, which property belongs to “the most important properties of the utilized PP”?
The authors thank the reviewer for the comment. For “the most important properties” the authors mean the most relevant thermo-mechanical properties. The caption of this table and the text have been modified accordingly.
BIBLIOGRAPHY
- Galvagnini, F.; Dorigato, A.; Fambri, L.; Fredi, G.; Pegoretti, A. Thermophysical Properties of Multifunctional Syntactic Foams Containing Phase Change Microcapsules for Thermal Energy Storage. Polymers (Basel). 2021, 13, 20.
Reviewer 2 Report
The study prepared a kind of novel polypropylene syntactic foams containing paraffin microcapsules. The rheological, morphological, thermal, and mechanical properties of the prepared materials were systematically investigated.
1. Figure 15 is interesting but a little hard to understand. According to the description, each red dot corresponds to one of the compositions in Table 9. It is recommended that author link the same compositions shown in Table 9 and Figure 15 by numbering or other means.
2. In theory, the enthalpy increases as the volume fraction of PCM increases. Why are the enthalpy values of H0-P20 and H20-P20 so different? And why the enthalpy of H20-P20 exceeds the theoretical value?
3. The dashed lines in Figure 7 make it difficult to distinguish lines of the same color scheme (H0-P20 and H0-P30, H20-P0 and H40-P0, H7P7 and H20-P20,). It is recommended to use line and symbol (as figure 3) or solid line (as figure 9).
4. Is the composite (PP+ compatibilizer+ HGMs+ PCM) simply a physical mix, or does it undergo a chemical reaction?
Author Response
Reviewer 2
- Figure 15 is interesting but a little hard to understand. According to the description, each red dot corresponds to one of the compositions in Table 9. It is recommended that author link the same compositions shown in Table 9 and Figure 15 by numbering or other means.
The authors thank the reviewer for the comment. As suggested, a specific number was assigned for each composition in Table 9, and reported in Figure 15.
- In theory, the enthalpy increases as the volume fraction of PCM increases. Why are the enthalpy values of H0-P20 and H20-P20 so different? And why the enthalpy of H20-P20 exceeds the theoretical value?
The authors thank the reviewer for the comment. It has to be considered that in DSC tests only one specimen was tested for each composition, and thus the difference detected in melting enthalpy values of H0-P20 and H20-P20 samples is justified by the experimental variability associated to these tests. Also the difference between the experimental and the theoretical enthalpy values of the H20-P20 foam, that is even smaller, can be explained in the same way.
- The dashed lines in Figure 7 make it difficult to distinguish lines of the same color scheme (H0-P20 and H0-P30, H20-P0 and H40-P0, H7P7 and H20-P20,). It is recommended to use line and symbol (as figure 3) or solid line (as figure 9).
The authors thank the reviewer for the comment. Figure 7 has been changed accordingly to the reviewer's suggestion.
- Is the composite (PP+ compatibilizer+ HGMs+ PCM) simply a physical mix, or does it undergo a chemical reaction?
The authors thank the reviewer for the comment. The authors are aware of the possibility of chemical reactions between the PP compatibilizer and the silanized HGMs, as already reported in some literature references [1–2]. However, a detailed analysis of this aspect is out of the main scope of this work, and it could be better investigated in future studies on this system.
BIBLIOGRAPHY
- Gogoi, R.; Manik, G. Development of thermally conductive and high-specific strength polypropylene composites for thermal management applications in automotive. Polym. Compos. 2021, 42, 1945.
- Bharath Kumar, B.R.; Doddamani, M.; Zeltmann, S.E.; Gupta, N.; Uzma; Gurupadu, S.; Sailaja, R.R.N. Effect of particle surface treatment and blending method on flexural properties of injection-molded cenosphere/HDPE syntactic foams. J. Mater. Sci. 2016, 51, 3793.
Round 2
Reviewer 2 Report
All the comments have been reviesed.